# Deep Link Entropy for Quantifying Edge Significance in Social Networks

**Seval Yurtcicek Ozaydin** [1],*[ID] **and Fatih Ozaydin** [2][ID]

1 Department of Social and Human Sciences, Tokyo Institute of Technology, 2-12-1 Ookayama, Meguro-ku, Tokyo 152-8552, Japan

2 Institute for International Strategy, Tokyo International University, 1-13-1 Matoba-kita, Kawagoe, Saitama 350-1197, Japan; fatih@tiu.ac.jp

* Correspondence: yurtcicek.s.aa@m.titech.ac.jp

**Abstract:** Through online political communications, fragmented groups appear around ideological lines, which might form echo chambers if the communications within like-minded groups are dominant over the communications among different-minded groups, potentially contributing to political polarization and extremism. The antidote is the interactions between individuals who constitute social bridges between different minded groups. Hence, exploring the significance of connections between the individuals of a network is a center of attraction especially for the global connectivity and diffusion in networks. Based on the divergence of probability distributions of pairs of nodes, Link Entropy (LE) is a recently proposed method outperforming the others in quantifying edge significance. In this work, considering that the adjacent nodes of the two nodes of an edge are also in charge in determining its significance, we propose the Deep Link Entropy (DLE) method for a more precise quantification through taking into account the uncertainty distributions of the adjacent nodes as well. We show experimentally that DLE significantly outperforms LE especially in large-scale complex network with several groups or communities. We believe our method contributes to not only online political communications but a wide range of fields from biology to quantum networks, where edge significance has an operational meaning.

**Keywords:** social networks; edge significance; fragmentation; echo chambers; group polarization; community detection; online political communications





## 1. Introduction

In a network with interacting communities consisting of nodes and edges each connecting a pair of nodes, some edges are of greater importance than others from a particular perspective. Consider a social network with communities fragmented along ideological lines, for example. Breaking of an edge connecting two nodes within a community and breaking of an edge connecting two nodes each belonging to another community lead to significantly different consequences in interrupting the communications between communities. The edge in the latter case is more *significant* than the edge in the former case in the sense that it is constituting a social bridge between two communities, enabling their communities' exposure to different ideological lines with a potential weakening of political echo-chambers and polarization. Therefore, quantifying the significance of nodes and edges of a natural or a human-made network is of crucial importance in various fields, especially in analyzing the connectivity and diffusion dynamics. Regarding the spread of computer viruses, for example, analyzing real-world data and through a dynamical model, Pastor-Satorras and Vespignani found that there is no epidemic threshold in scale-free networks that viruses can spread besides an arbitrary spreading rate [1]. Newman studied the structure of epidemic disease networks and showed that various models can be solved exactly on a wide variety of networks [2]. Based on real data in South Korea, Jo et al. has recently analyzed the social network of the spread of COVID-19 and found

that the size of the infection network could be significantly decreased if top nodes on the out-degree significantly are removed [3]. To reduce the energy consumption for wireless sensor networks, finding the redundant nodes based on a similarity measure, Wan et al. developed an efficient sleep scheduling mechanism [4] and Guruprakash et al. proposed a clustering approach [5]. Zio et al. designed multi-objective genetic algorithms to detect the edges of a realistic electrical network, which are critical for the structure and the flow [6].

Due to their social and political impacts, social networks are among the most popular structures in this context. Girvan and Newman's seminal work shed light on the community structure in social and biological networks [7]. To find the critical nodes of a sparse graph which can represent a social network with many nodes but a few connections, Zhou et al. developed a memetic search algorithm [8]. Based on the topological structure, particularly the cliques and paths, Yu et al. developed an edge-ranking algorithm [9].

Ranking the significance of edges is a core problem in the context of online political communications. Because the emergence of the Internet and social networks enabled faster and more efficient interactions between individuals. On the one hand, this is a great opportunity for societies that there is now an unprecedented chance for the diffusion of ideas among like- or different-minded groups, or communities. Through such a great opportunity for exposure to different ideologies, one might expect that the political polarization among like-minded groups weaken over time. On the other hand, however, there is also an unprecedented chance for gathering around the same ideological lines and for a deliberate enclave as well and this would inevitably contribute to forming and cementing echo chambers in fragmented groups [10]. That is, communications within communities become dominant over the communications among communities, paving the way for a debilitating polarization [11]. Along this line, Quattrociocchi et al. detected echo chambers on Facebook pages [12] and Adamic and Glance detected political polarization among political microblogs *(polblogs)* around the 2004 US presidential elections [13]. The *polblogs* network they studied became one of the standard networks to test various social network analysis approaches. Studying the anti-migrant sentiments on Twitter, Yurtcicek Ozaydin showed that even among like-minded groups reached consensus on a particular political issue, the reasons leading to the same ideological line can be sharply fragmented [14] and also detected deliberative enclaves [15,16] and sharp polarization [16,17] among fragmented groups on Twitter.

Communications between different-minded groups are enabled mainly by individuals who constitute a social bridge between the groups, each individual interacting with both groups. Regarding the global connectivity and network diffusion and especially for weakening echo chambers and political polarization, the edges connecting such pairs of individuals are more significant than the edges connecting individuals who are in interaction within their groups only. Several methods were proposed to quantify the edge significance, such as edge betweenness centrality [7,18,19], k-path centrality [20], degree product [21] and bridgeness [22]. The recent Link Entropy (LE) proposal of Qian et al. which is based on the divergence of probability distributions of pairs of nodes, outperforms these measures [23]. To calculate the significance of an edge connecting the nodes $n_i$ and $n_j$, Qian et al. calculates the entropies $H(X_i)$ and $H(X_j)$ of the nodes based on the probabilities $X_i$ and $X_j$ that $n_i$ and $n_j$ belong to each possible group, respectively, and the Jensen–Shannon divergence $JSD(X_i \parallel X_j)$.

The objective of this work is to quantify the significance of edges of a network more precisely and we ask whether it is possible to extract more information from the structure of the network. To answer this question, we consider that two different scenarios where the other adjacent nodes of $n_i$ or $n_j$, have a small, or a great uncertainty of belonging to one group might lead to significantly different results. Finding such a difference in the results would help shrinking the gap between the intrinsic information of a network's structure and our knowledge on it, contributing to the literature. Hence, aiming to provide a more precise edge significance quantification approach, we propose to improve the LE approach by taking into account also the entropies of the adjacent nodes of $n_i$ and $n_j$ and we call our

approach as Deep Link Entropy (DLE). Denoting the sum of entropic functions of node $n_i$ with $D(n_i)$ and the Link Entropy of the edge connecting the nodes $n_i$ and $n_j$ with $LE_{ij}$, DLE can be considered roughly as $DLE_{ij} = LE_{ij} + \chi[D(n_i) + D(n_j)]$, where $\chi$ is a small factor to be determined for each unique network. Note that, failing to find a non-zero $\chi$ for a given network, DLE reduces to LE by setting $\chi = 0$. Hence, one can safely claim that DLE performs at least as good at LE. Nevertheless, we were able to find a non-zero $\chi$ easily in our experiments on three typical networks, namely (*i*) the network presented by Wang et al. [24] for discovering groups (communities) in a network, (*ii*) Zachary's Karate Club Network [25] and (*iii*) American College Football Network [7].

Considering the edge-significance within the same context of Ref. [23], i.e., the global connectivity and diffusion dynamics, our tests for evaluating the performance of DLE follow the popular approach also used by Qian et al. [23] in their LE approach. We remove the edges one by one in descending order of significance found by each approach and in each step dividing the number of nodes in the largest component by the total number of nodes in the entire network, usually denoted by $R_{GC}$. Until the first disintegration, $R_{GC} = 1$ and as all the edges are removed, $R_{GC} \to 0$ for large $n$, number of nodes. Hence, observing the decrease in $R_{GC}$, it is straightforward to compare the performance of the approaches in consideration, i.e., the approach yielding a faster decrease of $R_{GC}$ towards zero performs better in quantifying the significance of edges.

## 2. Materials and Methods

In order to design a quantification strategy for the edges of a network, the first step is to discover the communities of the network, which is a crucial problem in a wide range of disciplines from sociology to biology and computer science. Fortunato has provided a comprehensive review of the algorithms in the literature to address this problem [26].

We choose to apply one of the most successful methods, namely the non-negative matrix factorization (NMF) method [24,27,28] to discover the communities. Note that the main purpose of this work is to quantify the significance of edges for global connectivity and diffusion dynamics. Detecting communities and nodes' community memberships is a preliminary step of the algorithm. Next, we quantify the edges according to our DLE approach, as detailed below.

To test the performance of our approach by comparing it to the original LE [23], in addition to the same community discovery algorithm, we follow exactly the same strategy to remove the most significant edge from the network and calculate $R_{GC}$, the fraction of nodes of the largest connected group. However, removing each edge results in a new network that might possibly have a different community structure and a new ranking of edge significance. Hence, before removing the next most significant edge in each step, we first apply NMF to discover the communities again and then quantify and rank the significance of the existing edges. As illustrated in the flowchart in Figure 1, we repeat this *update the ranking and remove the currently most significant edge* procedure until there is no edge left in the network.

We consider an undirected graph $\mathcal{G}$ consisting of $k$ unconnected communities, each of size $p_1, p_2, \ldots p_k$, where the weights of the edges in each community is $s_i$. $\mathbf{S}_i$ representing a $p_i \times p_i$ matrix (with elements $s_i$), $\mathbf{X}$ representing the cluster membership, $\mathbf{G}$ representing the adjacency matrix of graph $\mathcal{G}$, can be factorized as $\mathbf{G} = \mathbf{X}\mathbf{S}\mathbf{X}^\top$ where

$$\mathbf{G} = \begin{bmatrix} \mathbf{S}_1 & 0 & \ldots & 0 \\ 0 & \mathbf{S}_2 & \ldots & 0 \\ \vdots & \vdots & \ddots & \vdots \\ 0 & 0 & \ldots & \mathbf{S}_k \end{bmatrix}, \quad \mathbf{X} = \begin{bmatrix} 1 & 0 & \ldots & 0 \\ \vdots & 0 & \ldots & 0 \\ 1 & 0 & \ldots & 0 \\ 0 & 1 & \ldots & 0 \\ \vdots & \vdots & \ddots & \vdots \\ 0 & 0 & \ldots & 1 \end{bmatrix}, \tag{1}$$

and

$$\mathbf{S} = \begin{bmatrix} s_1 & 0 & \dots & 0 \\ 0 & s_2 & \dots & 0 \\ \vdots & \vdots & \ddots & \vdots \\ 0 & 0 & \dots & s_k \end{bmatrix}. \tag{2}$$

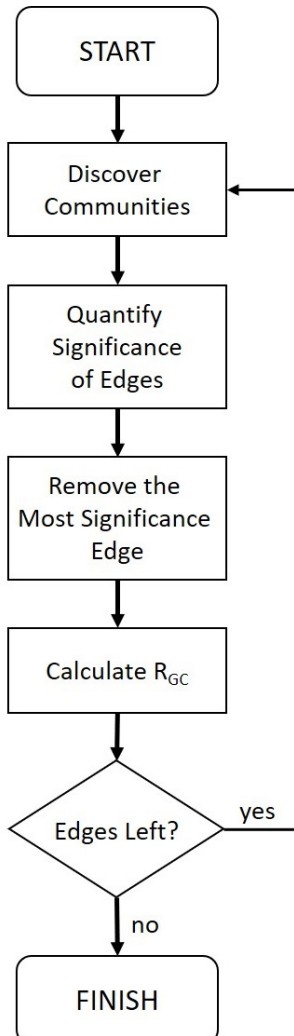

**Figure 1.** Flowchart of the present strategy, where the communities are discovered and the significance of edges are ranked in every step for removing the *currently* most significant edge each time, until all the edges are removed. In each step, $R_{GC}$, the fraction of nodes of the largest component is calculated for analyzing the performance of the approaches for quantifying the edge significance.

Here, diagonality of **G** and orthogonality of the columns of $X$ is due to unconnectedness of the communities. However in real applications, especially in social networks, communities are connected through some individuals, who are connected also to the individuals of some other networks. Hence, rather than the simple factorization, NMF can be used for discovering the communities by minimizing the loss

$$\min_{\mathbf{X} \geq 0,\ \mathbf{S} \geq 0} L(\mathbf{G}, \mathbf{XSX}^\top) \tag{3}$$

where the loss function can be defined as

$$L(A, B) = \|A - B\|_F^2 = \sum_{ij} (A_{ij} - B_{ij})^2. \tag{4}$$

Because we consider $\mathcal{G}$ as an undirected graph, **G** and therefore **S** are symmetric, so that the problem can be reduced to finding $X$ in the minimization

$$\min_{\mathbf{X} \geq 0} \left\| \mathbf{G} - \mathbf{X}\mathbf{X}^\top \right\|_F^2 \tag{5}$$

with $\mathbf{X} \leftarrow \mathbf{X}\mathbf{S}^{1/2}$, which can be done through the iteration [29].

$$\mathbf{X}_{ik} \leftarrow \mathbf{X}_{ik} \left( \frac{1}{2} + \frac{(\mathbf{G}\mathbf{X})_{ik}}{(2\mathbf{X}\mathbf{X}^\top \mathbf{X})_{ik}} \right). \tag{6}$$

Finally, normalizing the rows of **X**, each element $\mathbf{X}_{ik}$ gives us the probability that $i$-th node is a member of $k$-th community.

In the simulations, **X** is initially taken as a random matrix, which potentially affects the results. However, we update the ranking of the significance of edges in each step (i.e., each removal) and this is achieved by a new random $X$ matrix. Hence, -plausibly similar to a random walk process-, including not a single but too many initially random $X$ matrices, on the average, the effect of the randomness is minimized. What is more, the minimization in Equation (5) through the iteration in Equation (6) is executed a thousand times. Hence, though starting with a random matrix with elements ranging from 0 to 1 each time, **X** converges to the same result with insignificant differences in the elements. We confirmed this by repeating the simulations sufficiently many times and observing that although the results (curves of the plots) are not exactly the same in each run, the difference between the results of any two runs is practically insignificant, not affecting the overall results, i.e., the gap between DLE and LE does not shrink or expand significantly due to the randomness of $X$ matrix.

Having discovered the communities and memberships, the significance of each edge connecting the nodes $i$ and $j$ can be found by the Link Entropy (LE) [23] as

$$LE_{ij} = \frac{(H(\mathbf{X_i})) + (H(\mathbf{X_j}))/2 + JSD(\mathbf{X}_i \| \mathbf{X}_j)}{2} \tag{7}$$

through information entropy

$$H(\mathbf{X_i}) = -\sum_{k=1}^{K} x_{ik} \log x_{ik} \tag{8}$$

and Jensen–Shannon divergence for $M = \frac{1}{2}(\mathbf{X}_i + \mathbf{X}_j)$

$$JSD(\mathbf{X}_i \| \mathbf{X}_j) = \frac{D(\mathbf{X}_i \| \mathbf{M}) + D(\mathbf{X}_j \| \mathbf{M})}{2} \tag{9}$$

with

$$D(\mathbf{X}_i \| \mathbf{M}) = \sum_{k=1}^{K} x_{ik} \log \frac{x_{ik}}{m_k}, \tag{10}$$

whereas Deep Link Entropy (DLE) approach proposed in this work takes into account also the entropies $\mathbf{X}_{i,\text{adj}}$ and $\mathbf{X}_{j,\text{adj}}$ of the adjacent nodes of $n_i$ and $n_j$ of a network of $N$ nodes, respectively, as

$$DLE_{ij} = \frac{(H(\mathbf{X_i})) + (H(\mathbf{X_j}))/2 + JSD(\mathbf{X}_i \| \mathbf{X}_j)}{2} + \chi \left( \sum_{i \neq j}^{N} H(\mathbf{X}_{i,\text{adj}}) + \sum_{j \neq i}^{N} H(\mathbf{X}_{j,\text{adj}}) \right), \tag{11}$$

where $\chi$ is a parameter to be determined for achieving the highest performance in quantifying the significance of edges of a given network. Note that the extra terms are not added

arbitrarily to achieve a better performance, but they appear as a result of our theory that the entropies of the adjacent nodes of $n_i$ and $n_j$ affect the significance of the $e_{i-j}$ as well.

## 3. Results

Following the flowchart illustrated in Figure 1 and the approach presented in the Methods section, we rank the significance of edges of various networks according to LE and DLE and removing the most significant edge in each step, we test the performance of these two approaches in detecting the faster disintegration.

Because the basic improvement of DLE over LE is to take into account the adjacent nodes as well, if the network consists of more than two groups (or communities), there appears the chance that the adjacent nodes might belong to some other groups, which leads to the prediction that DLE can perform significantly better than LE particularly in multi-group networks, regardless of our assumption on the number of groups in the simulation.

We test our approach on three typical networks, the first being the one presented by Wang et al. [24] in their community detecting algorithm. The second and third networks are chosen to be popular benchmark networks in the literature, which are relatively small- and large-scale networks, respectively, so that we can measure the performance of the approach within a wide range. The first network consists of four interacting groups (communities). Because Qian et al. claimed that no matter how many groups the network consists of, considering only two groups perform slightly better and also the simulation runs faster than considering the actual number of groups, we quantify and rank the edges considering that there are two groups ($k = 2$), first, and then four groups ($k = 4$) in the network.

The adjacency graph of the network in consideration is given in Figure 2. We run the simulations for LE and DLE for $k = 2$ and $k = 4$. Removing the most significant edge according to LE and according to DLE, re-discovering the groups and re-quantifying the edges in each step, we find that DLE significantly outperforms LE as presented in Figure 3.

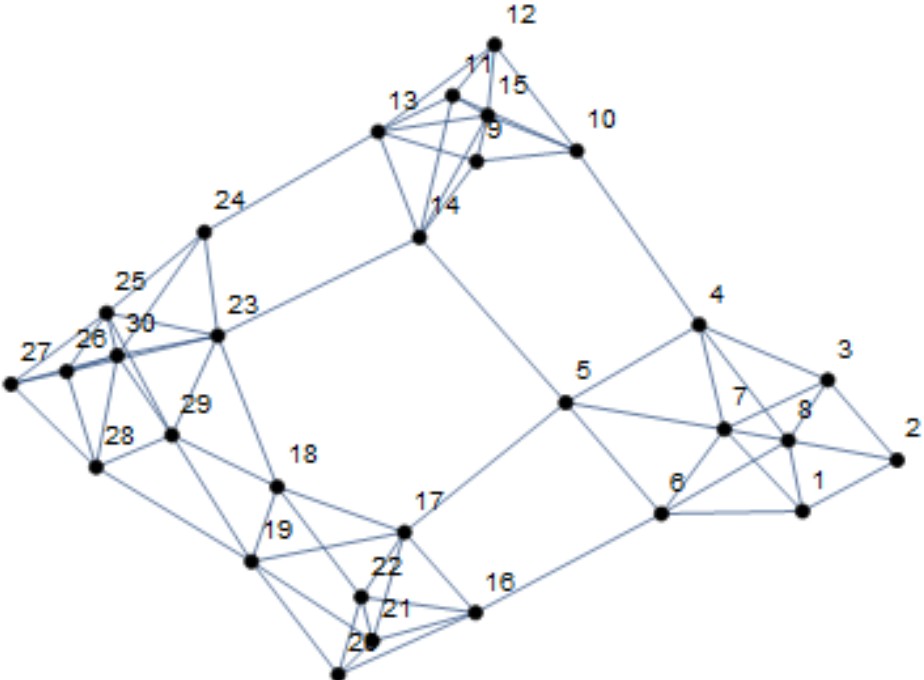

**Figure 2.** Adjacency graph of the network presented in the community discovery work of Wang et al. [24], consisting of four communities with overlapping nodes such as $n_5$ and $n_6$.

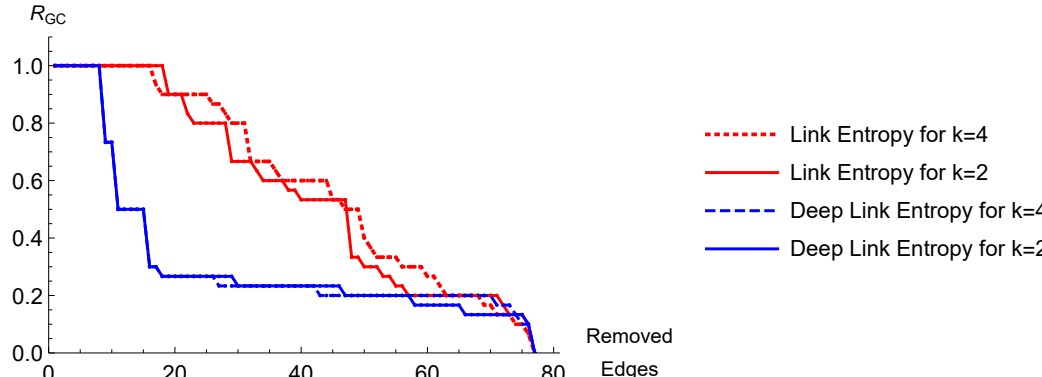

**Figure 3.** Testing Deep Link Entropy (DLE) on the network presented by Wang et al. [24] (adjacency graph illustrated in Figure 2) showing the fraction of nodes of the largest connected group, $R_{GC}$, as a result of removing the most significant edge in each step after discovering the communities and updating the edge significance ranking. DLE with $\chi = -0.12$ outperforms Link Entropy (LE), i.e., the best strategy so far.

Next, we analyze the Zachary's Karate Club network [25] with the adjacency graph presented in Figure 4, which is usually considered to be consisting of two groups. As shown in Figure 5, for the same factor of adjacent nodes, i.e., $\chi = -0.12$, we find that while at first (until around 20 edges are removed) LE performs better, DLE starts and keeps performing better than LE for the rest of the edge removal process.

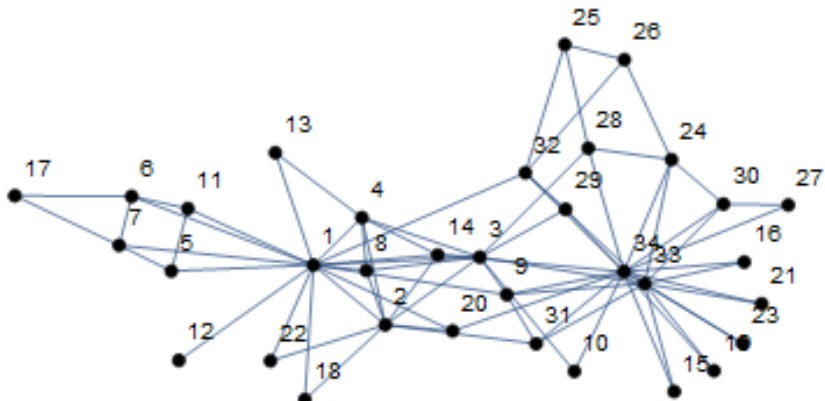

**Figure 4.** Adjacency graph of Zachary's Karate Club network [25].

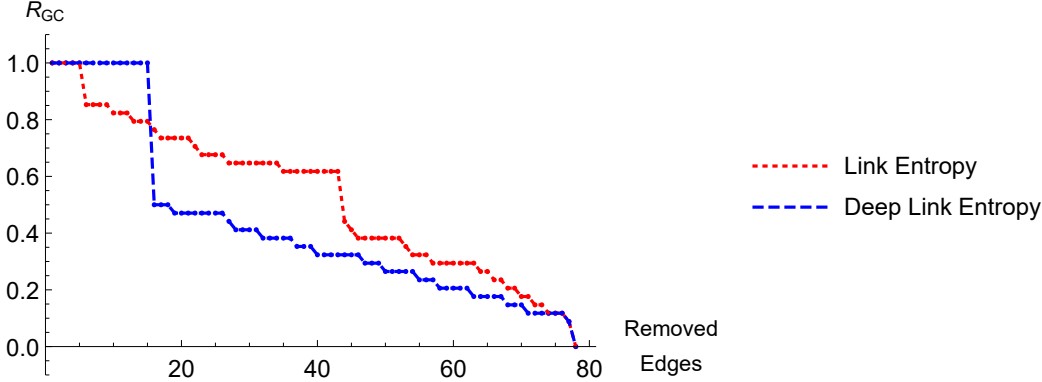

**Figure 5.** Testing Deep Link Entropy (DLE) on Zachary's Karate Club network [25] (adjacency graph illustrated in Figure 4) showing the fraction of nodes of the largest connected group, $R_{GC}$, as a result of removing the most significant edge in each step after discovering the communities and updating the edge significance ranking. Being outperformed in the first quarter of the process, DLE with $\chi = -0.12$ outperforms LE in the rest of the process.

Having analyzed two small-scale networks consisting of four and two groups, respectively, we continue with the main target of the present approach, i.e., a large-scale complex network consisting of several groups (communities). We take the American Football Network as an exemplary network, which has 115 nodes and 613 edges as shown in Figure 6. In the literature, different number of communities are detected by various strategies. Nevertheless, without loss of generalization and for shorter runtime of the simulation, following Qian et al., we consider two groups for testing DLE. As can be seen in Figure 7, we find that DLE performs significantly better than LE.

Because, in such a network, the uncertainty of adjacent nodes to belong to some other communities tends to be higher, that the adjacent nodes which we add to the original LE, become more important.

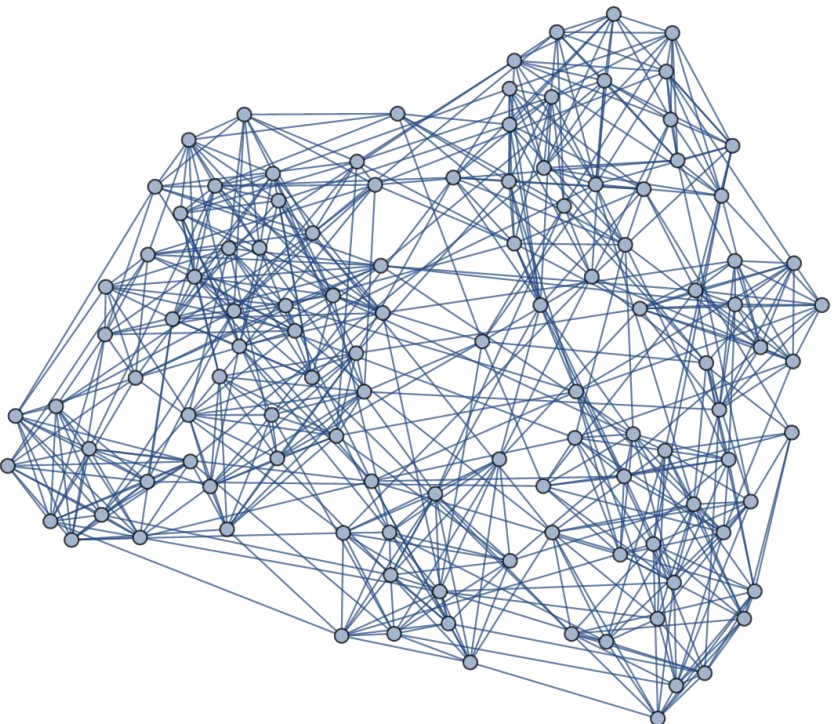

**Figure 6.** Adjacency graph of the American Football network with 115 nodes and 613 edges [7].

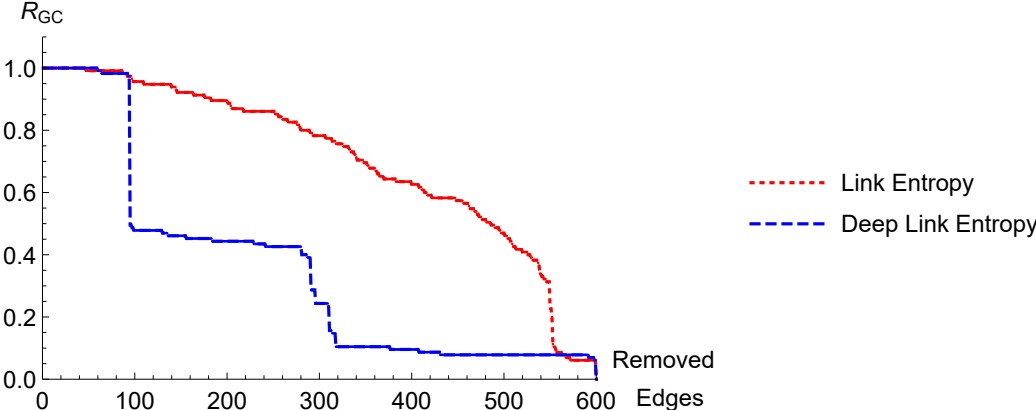

**Figure 7.** Testing Deep Link Entropy (DLE) on the American Football network [7] (adjacency graph illustrated in Figure 6) showing the fraction of nodes of the largest connected group, $R_{GC}$, as a result of removing the most significant edge in each step after discovering the communities and updating the edge significance ranking. DLE with $\chi = -0.12$ significantly outperforms LE in the large-scale complex network with several connected communities.

## 4. Discussion

An important and interesting point in our approach is determining the coefficient of the adjacent nodes, $\chi$. We anticipate possibly a different and even unique $\chi$ for each network for achieving the highest performance. Hence, we state an interesting open problem for future research to develop a theory for extracting the optimum value of $\chi$ directly from the network structure. Without extracting the value in a theoretical basis, however, it is possible to use optimization techniques to determine $\chi$. Nevertheless, we were able to find $\chi$ values easily such that DLE can outperform LE. On the first hand, we obtained our results for all the networks for a single value, i.e., $\chi = -0.12$. On the other hand, failing to find a non-zero $\chi$ for a given network, setting $\chi = 0$, the present DLE approach reduces to LE. Because the main purpose of the present work is to improve the LE approach by taking the adjacent nodes into account, we leave the open questions regarding $\chi$, together with other benchmarking networks such as dolphins or polblogs networks for future research. Note that the results we found herein regarding the LE approach might look slightly different than the results of the original LE work, due to the scaling of the figures and to the update mechanism we employ.

A crucial result of the present work is that according to our strategy, more than a continuous and smooth decrease, $R_{GC}$ exhibits sudden drops. In the former case, one node or a few nodes are continuously disintegrated from the overall network. However in the latter case, following the removal of edges between groups (with a nearly constant $R_{GC}$), as the last connecting edge is removed, one group completely disintegrates from the rest of the network, which we probe by a sudden drop of $R_{GC}$. Both cases have slightly different implications from the perspectives of individuals or groups disintegrating from the society, respectively.

Regarding political communications in online social networks in particular, through removing the edge quantified as the most significant in each step, our proposal disintegrates the groups from each other and from the society. This means that our approach successfully detects the edges constituting online bridges between groups which are fragmented along ideological lines. Beyond the problem of analyzing whether online political communications reinforce or weaken the echo chambers among fragmented groups and the group polarization [10–15], achieving a high performance edge significance quantification, we believe our work contributes to the field by putting forward an operational framework. That is, isolation through disintegrating from other different-minded groups and eventually from the society, echo chambers are formed and cemented within ideological groups, posing a threat to democracy. Hence, in order to strengthen the interaction between groups, detecting edges bridging the groups paves the way for social engineering towards an advanced democracy. Social network sites (SNS) are already implementing advanced algorithms for detecting accounts to suggest others to connect, i.e., to *follow* or to *become friends* . Their algorithms are plausibly targeting to increase the network traffic for maximizing their profit. For social engineering towards an advanced democracy—possibly through regulations—SNS could consider updating their algorithms to consider the global connectivity and diffusion of ideologies among communities. We conceptualize two sets of actions, (i) to keep existing connections, and (ii) to create new connections.

For the first set, considering the fact that nodes are disconnecting from others (through unfollowing, or unfriending) time to time due to a decrease in interactions, actions such as bringing the posts of one particular node to the timeline of another particular node more frequently can help increasing the interactions, hence, keeping the existing connections. Here, the key question is which pair of nodes, i.e., which edge to protect. Looking at the adjacency graph of network in Figure 2 suggests that unlike $e_{7-8}$, $e_{4-10}$ is a bridging edge between two fragmented communities. So, it would be more effective to protect $e_{4-10}$ than to protect $e_{7-8}$ for keeping those two communities connected and the increased interactions would contribute to weakening the echo-chambers and polarization within each community. That is, $e_{4-10}$ is expected to be more significant than $e_{7-8}$. Hence, a more precise quantification of edge significance in a given complex social network from the

perspective of global connectivity and diffusion of ideologies would have a positive impact on political communications, indicating the link between our results to the objective of this work.

For the second set, the key question is to pick which two nodes to suggest connecting. In the same network, a new edge $e_{14-17}$, or $e_{5-23}$ would be more effective than a new edge $e_{27-29}$. Being beyond the present work, we consider another method for determining new edges to suggest. For a given network, set of pairs of unconnected nodes (corresponding to elements in the adjacency matrix which are equal to 0), is constructed. One pair of nodes is picked and it is considered that they are connected (setting the corresponding element of adjacency matrix to 1). Running the edge significance algorithm, significance of this new edge is quantified. Repeating this step for every pair in the set, significance of each candidate edge is determined and SNS can suggest pairs of nodes with greatest edge significance to connect, so that the chance to create bridges between fragmented communities is increased.

In addition to social networks, our approach can be used for detecting the significant, or critical edges in a wide range of applications from biology to electrical networks. Moreover, considering von Neumann entropy of quantum particles, i.e., the nodes and considering the non-classical correlations between the particles as the edges, our approach can be adopted to quantum domain. In quantum networks based on multipartite entanglement, significance of edges are identical in typical states such as maximally entangled GHZ or W states [30,31]. However, for states with more than three particles, generic graph (or cluster) states emerge, enabling measurement-based one-way quantum computation [32], where groups of particles realizing the logic gates of the abstract gate-model can be considered as the *communities* of the network. In such generic networks, the edges, hence their significance are not identical. Hence, considering the scenario with a limited resource to protect only a set of edges against decoherence, or to apply error-correction operations in a quantum network, the edge-significance quantification problem becomes crucial for selecting the edges to act on. Hence, quantum extensions of our approach can find applications in entanglement percolation [33] and one-way quantum computers [32].

The extra terms in Equation (11) we added to design our DLE method based on LE method are found not by a brute force attack to provide an improved performance, but rather following our hypothesis that the uncertainties of adjacent nodes of $n_i$ and $n_j$ of belonging to each possible group contributes to quantifying the significance of edge $e_{i-j}$. Only the coefficient $\chi$ is to be determined which we easily did in our experiments. The drawback of adding these extra terms is the increasing computational complexity. Regardless of number of nodes in a given network, for each edge, LE computes four entropic functions. However, if $n_i$ has $I-1$ adjacent nodes (excluding $n_j$) and $n_j$ has $J-1$ adjacent nodes (excluding $n_i$) DLE computes additional $I+J-2$ entropic functions for each edge. These additional computations did not significantly increase the run time for networks with around 150 nodes and 1000 nodes on a state-of-art computer with 8th generation Core i9 Intel CPU, but it might start taking a considerable run time for social networks with thousands or even millions of nodes and edges, indicating a limitation of our method. However, emerging quantum computers are promising to perform computations significantly more efficient than classical computers that it would be interesting to calculate the quantum computational complexity of the present and similar graph problems.

A basic limitation of not only DLE but also LE and other methods for quantifying edge significance based on network structure (adjacency graph) only is that each node is assumed to be identical in behavior. Considering the characteristics and activities of nodes, analyzing for example how polarizing or uniting contents are posted as well as the frequency of posts and reactions would provide a better edge significance quantification and a better understanding of political polarization and dynamics of echo-chambers in fragmented communities through diffusion simulations.

## 5. Conclusions

In quantifying the significance of edges for maintaining the global connectivity and diffusion dynamics in complex networks consisting of several communities, though not taking into account the impact of adjacent nodes of the two nodes constituting that edge, the Link Entropy (LE) approach has been the most successful so far. In this work, we have proposed the so-called Deep Link Entropy (DLE) approach as a significant improvement over LE, reflecting the nature of complex networks, in particular the distinction whether those adjacent nodes *bridging* the communities or not. That is, we take into account the entropies of the adjacent nodes, indicating the certainty of belonging to either one or more communities. Through numerical simulations of basic types of the benchmark networks in the literature, we have shown that the improvement proposed in our approach yields significantly better results in quantifying edge significance in complex networks, especially consisting of several interacting communities. We discussed the social and political aspects of our work and possible extensions of it to several fields including quantum networks.

**Author Contributions:** Conceptualization, S.Y.O.; methodology, S.Y.O. and F.O.; software, F.O.; validation, S.Y.O. and F.O.; formal analysis, S.Y.O. and F.O.; writing—original draft preparation, S.Y.O. and F.O.; writing—review and editing, S.Y.O. and F.O.; visualization, S.Y.O. and F.O. All authors have read and agreed to the published version of the manuscript.

**Funding:** This research was funded by Personal Research Fund of Tokyo International University. The APC was funded by Personal Research Fund of Tokyo International University. S.Y.O. acknowledges Tokyo Institute of Technology Tsubame Scholarship.

**Institutional Review Board Statement:** Not applicable.

**Informed Consent Statement:** Not applicable.

**Data Availability Statement:** Not applicable.

**Acknowledgments:** S.Y.O. thanks to Ryosuke Nishida for fruitful discussions.

**Conflicts of Interest:** The authors declare no conflict of interest.

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
