# Peer review of "Deep Link Entropy for Quantifying Edge Significance in Social Networks"

_applsci, doi:10.3390/app112311182_

Round 1

Reviewer 1 Report

Thank you for the opportunity to review the article “Deep Link Entropy for Quantifying Edge Significance in Social Networks”. This is an engaging paper that studies the problem of the power and edge significance in a network. The findings are related to the social networks structure and the use of Deep Link Entropy method, and as the authors suggest, this method reflects better the nature of complex networks and of the relationships between the edges.

Overall, the subject is corrected attributed to this section of the journal “Computing and Artificial Intelligence” in regard to the aim of the study:  to elaborate a quantification strategy for the edges of a network.

However, there are some issues that should be addressed in this revision that I consider to be minor. First of all, I recommend the authors to take into consideration some corrections about the structure and the content of the article:

  1. The use of the syntagm” the other hand” suggest that there is something on the first hand, but not every time “the first hand” is specified (lines 110, 165)].
  2. Some aspects are not well developed or argued within the article, as:
    • The objectives of the study
    • Is there a gap of knowledge regarding the subject?
    • Why choose those three typical networks?
    • Why the non-negative matrix factorization method is one of the most successful methods to discover communities?
    • The impact on political communication
  3. As the authors mentioned some studies regarding, I also recommend using this article: Jo, W., Chang, D., You, M. et al. (2021) A social network analysis of the spread of COVID-19 in South Korea and policy implications. Sci Rep 11, 8581. https://doi.org/10.1038/s41598-021-87837-0

A major issue of this paper that I need to express is the following:

  1. The authors should describe better the limits of the research. I’ve noted they suggest further research, but no limitation is presented.
  2. Also, the objectives of the article should be clearly mentioned, and the results must be linked to these objectives.

Reviewer 2 Report

The main purpose of the paper is not clear. Community detection? Analyzing the role of each node? Nodes playing a specific role in community detection? Determining the coefficient of the adjacent nodes? They should focus on the problem addressed, and provide the corresponding background. •I miss an in deep literature review. Some of the classic method on measuring the importance of the communication are not even mentioned. Nevertheless, those works set the basis of the problems the authors address. See for example Freeman 1977 and 1979 or Fortunato 2010. •I miss a more detailed section of materials and methods. •What about the simulation process of the matrix X? It should be detailed explained, otherwise it is not possible to know the structure of the matrix over which the method is tested. •The experiment is really poor. Just two benchmark networks are considered (Zachary’s Karate Club and American College Football Network. This scarce test is not enough to claim the goodness of the proposedmethod. Moreover, they just focus on networks with k= 2 or k= 4 groups; in my humble opinion, other assumptions should be also tested. They do not provide results comparing with other existing methods; why they can affirm their proposal is good enough? •The authors use the work of Qian et al. as basis. I miss an explanation on this method before explaining their novelty. •The authors claim “we have proposed the the so-called Deep Link Entropy(DLE) approach as a significant improvement over LE”. However, they do not demonstrate this. •I miss the political and social discussion that the authors claim to have done. I agree it should be really interesting to analyze the role of the individuals in those contexts; this analysis would improve the quality andnovelty of the work. •I encourage the author to do a deep review of the punctuation and mis-prints.In general, I find the work is really poor. The authors try to provide a method to quantify edge significance in a network. However, they do not propose anything really new. The background is not clear, the methods and techniques are badly explained, and, the main problem, they do not detail the new proposal, the performance and results, as well as the comparison with other existing methods and evaluation of the technique. I also miss a proper discussion on the problem addressed. Then, I think it is too premature to publish the results

Reviewer 3 Report

The authors propose a new method for edge significance between nodes in different groups in networks. The proposed method extends an existing one by incorporating an entropy measure of the group memberships in the neighbourhoods of the nodes incident to an edge under study. The authors compare the performance of their method with the base method on example networks including the Karate club and a sports interaction network.

I recommend accepting the submission with a minor revision. Overall, I appreciate the theoretical derivation and discussion of the results. Moreover, the authors apply entropy, a concept with which a diverse audience is familiar, which enlarges the potential reach of the paper. However, the authors should address the following points in a minor revision to better fit the journal's scope, i.e. applied science.

  1. The paper lacks a precise definition of what a significant edge is. This is especially important given the diverse audience of the journal. For example, significance for a reader with a statistical background may mean an occurrence that is unlikely to be observed at random from a certain probabilistic model. But as far as I understand, this is not the approach of the proposed method. To rule out such confusion, the authors should clearly define what a significant edge is.
  2. The authors propose their method with possible applications to detect echo chambers in political interaction networks or significant edges in quantum networks. Here, I missed a discussion of how the chosen network approach relates to the information exchanged in the interactions underlying the edges. For example, for significant edges in a political network: how can we know that these correspond to a radicalized group of users instead of regular users who just intensely discuss a topic?
  • In general, the paper's language is sound, but a few typos remain, e.g.:
    • Line 142: "inf" instead of "in"
    • Line 179: "regarding LE approach" should mean "regarding the LE approach"
    • etc.

Reviewer 4 Report

The comments are in the attached file.

Round 2

Reviewer 2 Report

The improved the paper according with the suggestions done.